# Which Matters for Medical Utilization Equity under Universal Coverage: Insurance System, Region or SES

**DOI:** 10.3390/ijerph17114131

**Published:** 2020-06-10

**Authors:** Jiaoling Huang, Li Yuan, Hong Liang

**Affiliations:** 1School of Public Health, Shanghai Jiao Tong University School of Medicine, Shanghai 200025, China; huangjiaoling@shsmu.edu.cn; 2School of Social Development and Public Policy, Fudan University, Shanghai 200433, China; 3International Department, All China Youth Federation, Beijing 100051, China

**Keywords:** social medical insurance, utilization equity, influential factor, longitudinal analysis

## Abstract

Background: China has achieved universal coverage, with a higher rate of 95% medical insurance. However, huge inequalities are concealed under universal coverage. This article aims to explore the medical insurance utilization disparities over different insurance schemes, regions, and socioeconomic statuses (SES). Methods: This study was based on an open-access dataset in 2010, 2012, 2014, and 2016. A longitudinal analysis and separate logistic models were performed. Results: Urban Employee Basic Medical Insurance (UEBMI) members had an outstanding advantage in specialist visiting over those on the Urban Resident Basic Medical Insurance Scheme (URBMI) (OR = 0.607, *p* < 0.001) and New Cooperative Medical System (NCMS) (OR = 0.262, *p* < 0.001). However, in terms of a doctor visiting if a person is sick, the odds of patients in the NCMS receiving a visit were 55.1% ((OR = 1.551; *p* < 0.05) higher than those on the UEBMI. Compared with west China, the odds of those in the north-east and east were 2.1% (*p* > 0.05) and 97.2% (OR = 1.972; *p* < 0.001) higher for seeking medical treatment if sick, and 10.8% (OR = 0.892; *p* < 0.01) and 42.7% lower (OR = 0.573; *p* < 0.001) for a specialist visiting. In terms of SES, for each unit of increase in the Standard International Occupational Prestige Scale (SIOPS), the odds of seeking medical treatment decreased by 4.3% (OR = 0.958; *p* < 0.05), and the odds of a specialist visiting increased by 17.1% (OR = 1.171; *p* < 0.001) for each unit of the annual income logarithm. Conclusions: NCMS members and residents in west China were in a disadvantage status in terms of access to specialists, though had a higher probability of medical care if sick. SES variables were positively correlated with a specialist visiting consistently. We suggest a further focus on healthcare quality in the west and rural areas.

## 1. Introduction

During the past two decades, China has implemented a series of reforms to achieve the universal coverage of medical insurance. Since the foundation of China in 1949, a free universal medical insurance system was developed gradually, including the Labor Insurance Schemes (LIS) and the Government Employee Insurance Scheme (GIS) in urban areas and the cooperative medical scheme (CMS) in rural areas [1], in which urban residents and their families were financially supported by the government and state-owned units, while rural residents were placed into units of collective economy and provided with a basic level of healthcare protection [2]. Economic reform in the 1980s brought huge progress in economic development, which also caused numerous problems in healthcare programs, however. One of the significant reforms was introducing market incentives into healthcare institutions, resulting in profit-seeking behaviors in hospitals, such as prescribing expensive and unnecessary drugs and diagnostics [3]. The funds of the LIS, GIS, and CMS displayed a deficit gradually due to the rapid growth of medical expense and the huge waste of health resources [4]. The Chinese government was then forced to reform the free healthcare system and announced an ambitious universal coverage goal in the new round of healthcare reforms [5]. A new scheme, known as Urban Employee Basic Medical Insurance (UEBMI), was launched in 1998 [6], including public and private urban employment. However, their families were not covered. It was estimated that about 420 million urban residents were uninsured during that period [7]. To cope with this large uninsured population, the urban resident basic medical insurance scheme (URBMI) was launched in 2007 [8], mainly covering children, students, and the unemployed [9]. In rural areas, a new cooperative medical system (NCMS) was initiated in 2003, which was jointly funded by central and local governments and premiums [10]. By the end of 2011, more than 95% of the population was covered, compared with less than 50% in 2005 [11]. The ambitious goal of achieving universal coverage in medical insurance has been pushed on in an impressive coverage scale and speed [12].

However, huge inequalities were concealed under the universal coverage [13]. The criticism of it mainly focuses on three perspectives—i.e., system inequality, area inequality, and socioeconomic status (SES) inequality. One of the criticisms focuses on the variations in the system itself. Though almost all residents in China have been covered in different social medical insurance schemes, it is the household registration and employment status that determine the scheme rather than the residents’ own choices. Significant disparities were observed in the service benefit packages, including the drug provision, medical services, and reimbursement among different schemes [14]. A wide gap in premiums existed, with UEBMI having premiums 10 times higher than either the URBMI or NCMS [15]. For example, the range of drug coverage of UEBMI was 2510, whereas that of NCMS was 899 in the Hubei Province in 2012 [16]. Secondly, regional disparity has strengthened the system inequality. The current medical insurance system adopts a territorial management principal, which is mostly managed independently at the county level [17]. The system design of the same medical insurance system, including its insurance coverage, premium level, threshold, cap line, and reimbursement method, often varies from region to region [18,19]. For example, Shanghai medical insurance covered 4505 drugs in 2018 [20], while the number in Anhui Province was only 2837 [21]. Healthcare access and utilization inequality caused by regional disparity have been widely discussed and criticized, and the west, middle, and rural areas were always in a disadvantageous status [22,23,24]. Thirdly, SES-related medical insurance accessibility and affordability has also drawn great attention. Lower socio-economic class, often measured by income, education, and occupation, was found to be in a position of disadvantage in accessing medical resources [25,26]. The World Health Report also suggested that medical insurance utilization and costs were still very unequal across different sub-populations [27]. Such phenomena was also discovered in China. Xie E found that there was a pro-rich inequality in healthcare utilization; income’s contribution to inequality in healthcare use accounts for 0.13–0.2, and insurance enlarged the inequalities in healthcare use [28].

Such medical insurance inequalities in system, region, and SES must have an effect on medical utilization. Xiao N and colleagues argued that NCMS and URBMI—jointly financed by the central and province government and with a modest premium contribution from individual members—often did not adequately meet non-communicable disease (NCD) patients’ needs [29]. According to the WHO, universal health coverage is defined as ensuring that all people have access to needed health services of sufficient quality and also ensuring that they can afford the financial expense [30]. Equal access to healthcare is the primary target of medical insurance and also the sensitive index of affordability, which have been approved by current studies to different degrees [31,32,33]. However, few studies have focused on medical insurance utilization equity under universal coverage; most of them explored whether medical insurance could make a difference in healthcare utilization [34,35,36,37,38]. Some research paid attention to the system’s inequality in medical access; however, other important inequality factors, such as regional factors, have not be captured [39]. For example, Zhou ZL and colleagues explored the effect of UEBMI and URBMI on health utilization, and then compared the effect of the two medical insurances.

This paper aims to explore the medical insurance utilization equity after universal coverage has been achieved to examine the dominant influential inequality factors in medical utilization among insurance systems, regions, and SESs. Thus, our sample only focuses on insured individuals.

## 2. Materials and Methods

The data employed in this study was derived from China Family Panel Studies (CFPS), an annual longitudinal survey of Chinese communities, families, and individuals, which is performed by the Institute of Social Science Survey (ISSS) of Peking University. The CFPS is a nationally representative, biennial household survey, covering twenty-five provinces (excluding Hong Kong, Macao, Taiwan, Xinjiang, Tibet, Qinghai, Inner Mongolia, Ningxia, and Hainan), representing 94.5% of the total population in Mainland China [40]. CFPS conducted five waves of survey in 2010, 2012, 2014, 2016, and 2018, the last of which had been partly opened. The CFPS baseline cohort included 14,960 households in 2010. No new households entered the panel in the following wave, and the successful tracking rates were 85%, 89%, and 89% for the 2012, 2014, and 2016 follow-ups [41]. According to our research purpose, we kept the same insured residents each year, with the sample sizes of 23,069, 26,014, 27,620, and 26,204 in 2010, 2012, 2014, and 2016, respectively.

Dependent variables for this study included two aspects: (1) “have you sought medical treatment if you are sick within 2 weeks?” (1 = yes; 0 = no); (2) “Which institution do you prefer if you visit a doctor?” (1 = secondary or tertiary institution; 0 = primary institution). It is worth noting that only those who had become sick in the past two weeks were asked whether they had sought medical treatment. For the second dependent variable, the respondents were asked their usual preference when sick, and only those who preferred visiting a doctor (rather than self-medication) would be asked which institute they preferred. This is the main reason why the sample size differs from the overall. In China, hospitals are organized into a 3-tier system—i.e., tier-1, tier-2, and tier-3 hospitals [42], the latter two of which are commonly called large hospitals, while the first is officially called a community health service center (CHSC) in urban areas or a village clinic in rural areas. The CHSC set several stations in communities which belonged to the CHSC. A primary institution provided primary care by a general practitioner (GP), while large hospitals provided comprehensive medical treatment by specialists. A hierarchical medical system based on GP was set as a significant goal in healthcare reform in 2009; however, residents preferred to visit specialists in tier-2 and tier-3 hospitals rather than first-contact GPs, as more and higher quality resources (such as a health workforce and health technologies) were gathered in large hospitals, and primary-care counterparts were distrusted by consumers [43].

The independent variables were divided into three components: (1) Insurance system: “are you covered by any social medical insurance?” (1 = yes; 0 = no), “If so, which social medical insurance have you joined?” (1 = UEBMI; 2 = URBMI; 3 = NCMS; GIS = 4; 5 = Supplementary Medical Insurance (SMI)). Supplementary medical insurance was also included in the social medical insurance system, which was funded by enterprise. (2) Regional area. We adopted an official regional classification. According to the National Bureau of Statistics of China, China is divided into four areas—i.e., the eastern area (Beijing, Tianjin, Hebei, Shanghai, Jiangsu, Zhejiang, Fujian, Shandong, Guangdong, Hainan), the central area (Shanxi, Anhui, Jiangxi, Henan, Hubei, Hunan), the western area (Inner Mongolia, Guangxi, Chongqing, Sichuan, Guizhou, Yunnan, Tibet, Shanxi, Gansu, Qinghai, Ningxia, Xinjiang), and the north-eastern area (Liaoning, Jilin, Heilongjiang) [44]. (3) SES. Education, personal annual income, and Standard International Occupational Prestige Scale (Treiman’s SIOPS) were included. The occupational prestige indicator is formed by a list of occupations where respondents are asked to rate the popularity or approval of each occupation on the list, and then the percentage of approval for each occupation is tallied to obtain an occupational prestige score. In 1977, sociologist Donald Treiman, after comparing occupational prestige data from 60 countries, provided an international version of SIOPS, applicable to different countries, with scores ranging from 0 to 100 [45]. A logarithm was performed on income to ensure the normal distribution of the data, and we transformed the SIOPS to SIOPS/10, (per 10-unit increase) to be able to report a > 0% difference.

Control variables were the individual demographic characteristics, including gender (1 = male; 0 = female), age, household registration (1 = agricultural; 2 = non-agricultural; 3 = others), marital status (1 = never married; 2 = married; 3 = others), retired (1 = yes; 0 = no), NCD (1 = yes; 0 = no), and self-rated health (SRH) (1 = very unhealthy; 2 = unhealthy; 3 = relatively unhealthy; 4 = fair; 5 = healthy).

Population-averaged models were conducted for a longitudinal analysis on whether people sought medical treatment if sick within 2 weeks (Model-1 to Model-4) and whether they visit a specialist (Model-9 to Model-12). The data from four waves of the survey were mixed, and the standard errors were adjusted for clustering [46]. Taking the first dependent variable as an example, the demographic variables (Model-1), insurance schemes (Model-2), regional areas (Model-3), and SES variables (Model-4) were controlled step by step, which meant the effect of each factor could be observed explicitly. It is worth noting that, as the annual income and occupation variables are only asked for those who have a job currently, the sample size of Model-4/Model-12 will have a dramatic decrease, which also applies to Model-5 to Model-8, Model-12, and Model-13 to Model-16. Separate models were performed for each wave to capture the effect of the change in influential factors for medical treatment (Model-5 to Model-8) and specialist visiting in large hospitals (Model-13 to Model-16). Thus, we could observe all the coefficients each wave, and the changes in coefficients over waves could also be captured. All the analyses were performed with STATA version 13.0 (Stata Corp LP, Texas City, TX, USA). A significance level of 0.05 was employed for all the analyses.

## 3. Results

### 3.1. Characteristics of Study Participants

Table 1 indicates the socio-demographic statistics of the sample over waves, including the number (*n*), percentage (%), mean value (mean), and standard deviation (SD). The characteristic did not change much over time, as the respondents were tracked over years. Take 2016 as an example. Table 1 indicates that 13,145 (50.16%) respondents were male; the average age was 49.37 (± 16.05) and 18,043 were rural residents (73.99%), while the percentage of their urban counterparts was 25.92%. Indeed, 21,612 (82.48%) of the respondents were married; 13,413 (51.21%) of the respondents had graduated from primary school or below; 8370 (31.95%), 6061 (23.14%), 8365 (31.93%), and 3400 (12.98%) of residents lived in the west, center, east, and north-east of China, respectively; 5207 (19.87%) of the respondents were retired; 19,446 (74.71%), 3728 (14.28%), 2166 (8.30%), 624 (2.39%), and 134 (0.51%) of the respondents reported that they were covered by the NCMS, UEBMI, URBMI, Government Employee Insurance Scheme (GIS), and SMI, respectively (see Table 1).

### 3.2. Influential Factors in Medical Treatment for the Insured

Table 2 shows the pooled estimators of medical treatment, including four waves of data. Model-1 controlled demographic variables and showed that gender, age, household registration, marital status, being retired, NCD, and SRH were also significantly correlated with medical treatment for the insured. Model-2 to Model-4 examined the inequal factors of insurance system, region, and SES. Model-2 indicated that NCMS was in an advantageous position for obtaining medical treatment if sick. Specifically, the odds for NCMS were 55.1% (OR = 1.551; 95% CI: 1.368–1.758; *p* < 0.05) higher than for UEBMI. Model-3 showed that, compared with the west of China, the odds for the eastern and north-eastern areas were 10.8% (OR = 0.892; 95% CI: 0.827–0.963; *p* < 0.01) and 42.7% lower (OR = 0.573; 95% CI: 0.521–0.629; *p* < 0.001). Model-4 revealed that education and income were negatively correlated with medical treatment. Specifically, the odds for people who went to high school or had a bachelor’s degree were 79.7% (OR= 0.797; 95% CI: 0.665–0.954; *p* < 0.05) and 71.0% (OR = 0.710; 95% CI: 0.558–0.900; *p* < 0.01) of those whose education level was primary school or below. For each unit of increase in the logarithm of annual income, the odds decreased by 4.3% (OR = 0.958; 95% CI: 0.923–0.994; *p* < 0.05) on average (see Table 2).

Logistic regression models (Model-5 to Model-8) were performed to capture the changes in the odds ratios (ORs) of medical treatment behavior over years. The significant effects of age and NCD and SRH increased over years, while marriage lost its effect gradually. Unequal factors were observed to have significant changes. No significant disparity existed in 2010 among different insurance schemes. Disadvantageous medical treatment was captured for URBMI and SMI, the odds of which were only 61.1% (OR = 0.611, 95% CI: 0.389–0.961; *p* < 0.05) and 22.1% (OR = 0.221; 95% CI: 0.067–0.721; *p* < 0.05) of that of UEBMI. After 2014, a significantly higher probability was revealed for the NCMS-insured. The comparative OR (Ref. = UEMBI) increased from 1.606 (OR = 1.606; 95% CI: 1.116–2.313; *p* < 0.05) to 1.779 (OR = 1.179; 95% CI: 1.069–2.961; *p* < 0.05) from 2014 to 2016. Those in north-eastern China had a lower probability of seeking medical treatment if sick—i.e., the comparative ORs (Ref. = west) were 0.579 (OR = 0.579; 95% CI: 0.425–0.787; *p* < 0.001), 0.531 (OR = 0.531; 95% CI: 0.376–0.750; *p* < 0.001), 0.500 (OR = 95% CI: 0.375–0.666; *p* < 0.001), and 0.555 (OR = 0.555; 95% CI: 0.325–0.948; *p* < 0.01), respectively. The odds for those in central China were 77.4% (OR = 0.774; 95% CI: 0.600–0.999; *p* < 0.01) of those in the western area; however, such a significant difference disappeared over the years. Surprisingly, we did not observe significant coefficients for SES variables except for education. In 2010, the odds of residents with a bachelor’s degree were 37.2% (OR = 0.372; 95% CI: 0.230–0.603; *p* < 0.001) of that of their counterparts with primary school education or below; however, such a disparity within education disappeared over the years (see Table 3).

A longitudinal analysis was performed with four waves of data pooling. Model-9 showed that age, household registration, being retired, NCD, and SRH were significant variables. It is worth noting that the odds for non-agricultural household registration and others were 5.090 (OR = 5.090; 95% CI: 4.870–5.319; *p* < 0.001) and 2.467 (OR = 2.467; 95% CI: 1.450–4.195; *p* < 0.001) times lower than that of agricultural residents. Model-10 showed that, compared with UEBMI, the odds of URBMI, NCMS, and SMI were 39.3% (OR = 0.607; 95% CI: 0.563–0.654; *p* < 0.001), 73.8% (OR = 0.26295% CI: 0.243–0.281; *p* < 0.001), and 28.1% (OR = 0.719;95% CI: 0.623–0.830; *p* < 0.001) lower, indicating that the residents covered by UEBMI were more likely to visit specialists in large hospitals. In Model-11, compared with western China, the odds for the insured residents in the central, eastern, and north-eastern areas were 11.0% (OR = 0.890; 95% CI: 0.841–0.940; *p* < 0.001) lower, and 2.1% (OR = 1.021; 95% CI: 0.969–1.074; *p* > 0.05) and 97.2% (OR = 1.972; 95% CI: 1.850–2.101; *p* < 0.001) higher, respectively. Model-12 indicated that higher SES residents had an advantage in seeking higher quality medical resources. Specifically, compared with primary school or below graduates, the odds for graduates of middle school and high school and those with a bachelor’s degree and Master’s degree or higher were 22.6% (OR = 1.226; 95% CI: 1.116–1.347; *p* < 0.001), 61.8% (OR = 1.618; 95% CI: 1.439–1.818; *p* < 0.001), 119.6% (OR = 2.196; 95% CI: 1.894–2.547; *p* < 0.001), and 253.5% (OR = 3.535; 95% CI: 1.809–6.906; *p* < 0.001) higher. The odds of medical treatment increased on average by 17.1% (OR = 1.171; 95% CI: 1.137–1.205; *p* < 0.001) for each unit of increase in the logarithm of personal annual income (see Table 4).

Model-13 to Model-16 revealed the effect change of factors over the years. In terms of system inequality, the disparity between UEBMI and NCMS was narrowing. Specifically, the comparative ORs for NCMS (Ref. = UEBMI) were 0.371 (OR = 0.371; 95% CI: 0.212–0.649; *p* < 0.001), 0.474 (OR = 0.474; 95% CI: 0.377–0.597; *p* < 0.001), 0.432(OR = 0.432; 95% CI: 0.362–0.516; *p* < 0.001), 0.453 (OR = 0.453; 95% CI: 0.368–0.557; *p* < 0.001), presenting a growing trend. Regional factors had changed significantly; the insured residents in western area preferred large hospitals when seeking medical treatment in 2010, while the odds for central and eastern area residents were 70.4% (OR = 0.296; 95% CI: 0.201–0.433; *p* < 0.001) and 53.1% (OR= 0.469; 95% CI: 0.342–0.641; *p* < 0.001) lower in ORs. However, insured north-eastern residents tended to stay at a higher probability for visiting specialists than their counterparts in the west since 2012—i.e., the comparative ORs for the north-east were 1.935 (OR = 1.935; 95% CI: 1.569–2.386; *p* < 0.001), 2.037 (OR = 2.037; 95% CI: 1.727–2.401; *p* < 0.001), and 2.167 (OR = 2.167; 95% CI: 1.688–2.781; *p* < 0.001). We also found that the education-related disparity of specialist visiting significantly reduced after a small rise. In 2010, the odds of residents with a Master’s degree were 1.192 times (OR = 1.192; 95% CI: 0.140–10.093; *p* > 0.05) higher than that of primary school graduates, which had increased to 19.964 (OR = 19.964; 95% CI: 2.543- 156.667; *p* < 0.01) in 2012 and then dropped dramatically to 2.899 (OR = 2.899; 95% CI: 1.120–7.495; *p* < 0.05) in 2014 and 1.821 (OR = 1.821; 95% CI: 0.750–4.419; *p* > 0.05) in 2016. Personal annual income had a comparatively stable effect on specialist visiting; the ORs of which were 1.368 (OR = 1.368; 95% CI: 1.191–1.571; *p* < 0.001), 1.198 (OR = 1.198; 95% CI: 1.122–1.279; *p* < 0.001), 1.085 (OR = 1.085; 95% CI: 1.048–1.122; *p* < 0.001) and 1.315 (OR = 1.315; 95% CI: 1.199–1.441; *p* < 0.001), respectively, for each wave. The positive effect of occupational prestige became statistically significant in the last wave (OR = 1.087; 95% CI: 1.028–1.148; *p* < 0.001) (see Table 5).

## 4. Discussion

In contrast to current studies, we only focused on insured residents rather than all the population, and explored the inequality of medical treatment for the insured population. There are some interesting findings in this study which are distinguished from previous findings. As expected, we found that insurance system, region, and SES were all significant factors affecting equal medical utilization.

Consistent with previous studies, the residents covered by the UEBMI had an outstanding advantage in seeking quality healthcare resources compared to those covered by URBMI and NCMS [19,47]. For example, Wang and colleagues found that individuals with UEBMI had the highest healthcare costs and demonstrated the greatest effect of health insurance on healthcare utilization and expenses increases [48]. Niu and colleagues argued that the promotional effects on health service use differed across the insurance programs, with the NRCM and the URBMI showing comparable but lower impacts as compared with the UEBMI [17]. This could be caused by the varied reimbursement policy, in which UEBMI members had an advantage, with the lowest out-of-pocket (OOP) proportion. Besides, residents could only visit hospitals where their households are registered, which might also limit the specialist visiting. However, in terms of a doctor visiting if a person is sick, patients in the NCMS had a higher probability of a doctor visiting than their counterparts, even those in the UEBMI. According to Li and colleagues, the strengthening of primary healthcare in rural areas might have a positive effect on access to medical treatment for rural residents [49]. Liu et al. argued that the number and quality of health professionals at township health centers had a steady increase, and the amount of medical and public health services provided by township health centers had also increased significantly [50]. However, separate models in this study suggested that the disparities between the NCMS and UEBMI had decreased over years. The medical financial burden was still remarkably high for low-income rural residents in China due to high OOP payment, even with NCMS reimbursement [51]. It is necessary to further improve NCMS patients’ financial ability to access a specialist visiting, especially for inpatients and poor rural residents. It is suggested that service accessibility and affordability for vulnerable rural residents should be protected by modifying regressive financing in NCMS, and by providing extra financial aid and reimbursement from the government [51].

Medical treatment also varied by regional area. We found that people in the west areas were more likely to seek medical treatment if sick, though they had a lower probability of visiting specialists. Besides this, we noticed that insured residents in northeast and east China made greater use of quality medical resources than those in the central and western regions, and the disparities kept increasing. Recent studies focused on healthcare regional equality and obtained encouraging results. Zhang et al. conducted a questionnaire survey in eight cities in west China, and drew the conclusion that new reform initiated in 2009 had a better impact on the west area, especially in primary healthcare, stating “In community health comprehensive reform the changes are better in the west and middle parts than in the east part.” [52] Liu insisted that this reform has improved medical care utilization more for residents in poorer regions [53]. However, other studies found there was insufficient use of quality medical treatment for the west area. Sun and Luo evaluated the equality and efficiency of health resource allocation and health service utilization in China and found distinct regional disparities in healthcare resources and utilization. They found that people living in the eastern developed areas were more likely to utilize outpatient care, while their western counterparts were more likely to use inpatient care [54]. Thus, we inferred that medical insurance might have improved the access to medical treatment for the western residents gradually, but had not relieved the financial pressure of expert outpatient and hospitalization.

We found explicit evidence that SES was positively correlated with the medical utilization of the insured. Residents with a higher income had a higher probability of visiting specialists in large hospitals. Similar results were also found by other studies [55,56,57,58]. For example, Wang and colleagues investigated inequality in service utilization among the middle-aged and elderly in Gansu and Zhejiang provinces, and they found that income was the dominant factor in healthcare utilization inequality for outpatients in Gansu and Zhejiang [24]. A study conducted by Flatø and Zhang Richer pointed out that richer users were considerably more likely than the poor to seek care at hospitals rather than at clinics or health centers, and that the pro-rich inequality in the level of healthcare utilization was highly inequitable [59]. However, we found that education is negatively correlated with the doctor visiting if one is sick. We inferred that the higher-educated residents might have a better self-managed healthacare ability, which was echoed by the protocol study conducted by Baker et al. They found that participants with a better education had a lower healthcare use than their counterparts [60]. Self-management education was proved to be effective by numerous studies, and thus we suggest that it was a key strategy to promote self-management education to increase the efficient utilization of limited healthcare resources.

## 5. Limitation

There are several limitations of our study. First, we stated that it was a nation-wide sample which was representative of the population; however, we could not prove that the insured sample within the surveys was representative of the insured target population, as the characteristics of the insured population information are absent from current statistic yearbooks. Second, the CFPS survey adopted telephone interviewing to improve the response rate (the proportion of telephone interviews rose to 20% in 2016). Telephone interviews are not comparable to face-to-face interviews, which are thought to be a better way to collect higher quality data. Third, we ignored one special group of people who lived in the suburbs of urban areas but were covered in NCMS in this study. They are different from UEBMI/URBMI members, and also different from NCMS members. We will look into this group of people in future research.

## 6. Conclusions

Medical insurance system, region, and SES were all significant factors influencing medical utilization under universal coverage. Compared with other insurance schemes, NCMS members had a higher probability of obtaining medical treatment if sick, though they were less likely to visit specialists in large hospitals. Similarly, west China residents also had an advantage in seeking medical treatment, but a disadvantage in a specialist visiting. Except for education, the SES variables were positively correlated with medical utilization. We suggest a further focus on quality healthcare for the west and rural areas.

## Figures and Tables

**Table 1 ijerph-17-04131-t001:** Socio-demographic characteristics of the sample.

Variable	Category	2010	2012	2014	2016
Gender, *n*(%)	Male	11,790 (51.11%)	13,058 (50.20)	13,855 (50.16)	13,145 (50.16)
	Female	11,279 (48.89%)	12,956 (49.80)	13,765 (49.84)	13,059 (49.84)
Age, Mean (±SD)		45.56 (±15.74)	45.59 (±16.14)	47.71 (±16.19)	49.37 (±16.05)
Household Registration, *n*(%)	Rural	17,506 (75.89%)	19,805 (76.43%)	20,050 (74.70%)	18,043 (73.99%)
	Urban	5526 (23.95%)	6067 (23.41%)	6772 (25.23%)	6320 (25.92%)
	Others	37 (0.16%)	40 (0.15%)	17 (0.06%)	23 (0.09%)
Marital Status, *n*(%)	Single	2670 (11.58%)	3111 (11.96%)	2828 (10.24%)	2270 (8.66%)
	Married	18,954 (82.18%)	21,143 (81.29%)	22,562 (81.69%)	21,612 (82.48%)
	Others	1440 (6.24%)	1756 (6.75%)	2230 (8.07%)	2321 (8.86%)
Education, *n*(%)	Primary school or below	12,047 (52.24%)	13,632 (52.44%)	12,763 (48.77%)	13,413 (51.21%)
	Middle school	6857 (29.74%)	7289 (28.04%)	7559 (28.89%)	6921 (26.43%)
	High school	2824 (12.25%)	3362 (12.93%)	3781 (14.45%)	3478 (13.28%)
	Bachelor degree	1304 (5.66%)	1671 (6.43%)	2013 (7.69%)	2300 (8.78%)
	Master or higher	27 (0.12%)	40 (0.15%)	52 (0.20%)	78 (0.30%)
Region, *n*(%)	West	7257 (31.46%)	8429 (32.50%)	8746 (31.67%)	8370 (31.95%)
	Center	5378 (23.31%)	6183 (23.84%)	6552 (23.72%)	6061 (23.14%)
	East	7235 (31.36%)	8066 (31.10%)	8724 (31.59%)	8365 (31.93%)
	Northeast	3199 (13.87%)	3255 (12.55%)	3598 (13.03%)	3400 (12.98%)
Retired, *n*(%)	Yes	1741 (7.55%)	1924 (7.40%)	3808 (13.79%)	5207 (19.87%)
	No	21,328 (92.45%)	24,090 (92.60%)	23,812 (86.21%)	20,997 (80.13%)
Social Medical Insurance, *n*(%)	UEBMI	1965 (9.61%)	2984 (11.58%)	3553 (13.27%)	3728 (14.28%)
	URBMI	1544 (7.55%)	1870 (7.26%)	2298 (8.58%)	2166 (8.30%)
	NCMS	15,782 (77.18%)	19,850 (77.05%)	19,942 (74.47%)	19,446 (74.71%)
	GIS	1045 (5.11%)	947 (3.68%)	789 (2.95%)	624 (2.39%)
	SMI	111 (0.54%)	113 (0.44%)	198 (0.74%)	134 (0.51%)

Note: SD: standard deviation; UEBMI: Urban Employee Basic Medical Insurance; URBMI: Urban Resident Basic Medical Insurance; NCMS: New Cooperative Medical Scheme; GIS: Government Employee Insurance Scheme; SMI: Supplementary Medical Insurance.

**Table 2 ijerph-17-04131-t002:** Longitudinal analysis of seeking medical treatment within 2 weeks if sick.

Variable	Model-1			Model-2			Model-3			Model-4		
	B	SE	OR (95%CI)	B	SE	OR (95%CI)	B	SE	OR (95%CI)	B	SE	OR (95%CI)
Gender (Ref. = Female)	−0.107 ***	0.030	0.899 (0.847–0.954)	−0.100 **	0.031	0.905 (0.852–0961)	−0.116 ***	0.031	0.891 (0.838–0.946)	−0.048	0.057	0.953 (0.853–1.649)
Age	0.011 ***	0.001	1.011 (1.008–1.013)	0.010 ***	0.001	1.011 (1.008–1.013)	0.011 ***	0.001	1.011 (1.008–1.013)	0.008 ***	0.003	1.008 (1.003–1.014)
Household Registration (Ref. = Agricultural)												
Non-Agricultural	−0.404 ***	0.034	0.668 (0.625–0.713)	−0.080	0.057	0.923 (0.825–1.033)	−0.045	0.058	0.956 (0.852–1.070)	0.069	0.104	1.072 (0.875–1.313)
Others	0.138	0.486	1.148 (0.442–2.977)	0.047	0.519	1.048 (0.379–2.896)	0.104	0.560	1.109 (0.370–3.321)	0.100	0.977	1.105 (0.163–7.494)
Marital Status (Ref. = Never married)												
Married	0.304 ***	0.062	1.356 (1.200–1.531)	0.327 ***	0.065	1.386 (1.219–1.576)	0.334 ***	0.066	1.397 (1.228–1.589)	0.357 **	0.118	1.429 (1.134–1.800)
Others	0.151	0.084	1.163 (0.986–1.372)	0.174 *	0.087	1.190 (1.003–1.410)	0.192 *	0.087	1.212 (1.022–1.437)	0.480 **	0.170	1.615 (1.156–2.256)
Retired (Ref. = no)	0.263 ***	0.047	1.300 (1.187–1.425)	0.286 ***	0.047	1.331 (1.214–1.459)	0.277 ***	0.047	1.320 (1.203–1.448)	0.078	0.106	1.081 (0.878–1.332)
NCD (Ref. = no)	0.953 ***	0.035	2.594 (2.419–2.779)	0.956 ***	0.036	2.602 (2.425–2.791)	0.957 ***	0.036	2.604 (2.427–2.794)	0.928 ***	0.068	2.529 (2.214–2.888)
SRH (Ref.= Very Unhealthy)												
Unhealthy	−0.510 ***	0.041	0.600 (0.554–0.651)	−0.499 ***	0.041	0.607 (0.559–0.658)	−0.519 ***	0.041	0.595 (0.549–0.645)	−0.588 ***	0.090	0.555 (0.465–0.663)
Relatively Unhealthy	−0.677 ***	0.040	0.508 (0.470–0.549)	−0.658***	0.040	0.518 (0.479–0.560)	−0.676 ***	0.040	0.509 (0.469–0.550)	−0.703 ***	0.084	0.495 (0.420–0.583)
Fair	−0.669 ***	0.046	0.512 (0.468–0.560)	−0.653 ***	0.047	0.521 (0.475–0.570)	−0.658 ***	0.047	0.518 (0.473–0.568)	−0.661 ***	0.092	0.516 (0.431–0.617)
Healthy	−0.762 ***	0.058	0.467 (0.416–0.523)	−0.758 ***	0.060	0.469 (0.416–0.526)	−0.779 ***	0.060	0.459 (0.408–0.516)	−0.723 ***	0.108	0.485 (0.393–0.599)
Social Medical Insurance (Ref.= UEBMI)												
URBMI				0.083	0.063	1.087 (0.961–1.228)	0.062	0.063	1.064 (0.941–1.203)	−0.055	0.115	0.947 (0.756–1.186)
NCMS				0.439 ***	0.064	1.551 (1.368–1.758)	0.377 ***	0.065	1.458 (1.283–1.656)	0.367 **	0.114	1.443 (1.154–1.805)
GIS				0.100	0.084	1.106 (0.938–1.302)	0.105	0.084	1.111 (0.941–1.310)	0.130	0.128	1.139 (0.885–1.465)
SMI				−0.032	0.127	0.969 (0.756–1.242)	−0.052	0.127	0.949 (0.740–1.217)	−0.297	0.189	0.743 (0.513–1.075)
Area (Ref. = West)												
Center							−0.045	0.042	0.956 (0.881–1.038)	−0.144	0.077	0.866 (0.745–1.007)
East							−0.114 **	0.039	0.892 (0.827–0.963)	−0.081	0.070	0.922 (0.804–1.058)
North-East							−0.557 ***	0.048	0.573 (0.521–0.629)	−0.622 ***	0.089	0.537 (0.451–0.638)
Education (Ref. = Primary School or Below)												
Middle School										0.032	0.071	1.032 (0.898–1.186)
High School										−0.227 *	0.092	0.797 (0.665–0.954)
Bachelor’s Degree										−0.343 **	0.122	0.710 (0.558–0.900)
Master’s Degree										−0.291	0.383	0.748 (0.353–0.583)
Personal Annual Income										−0.043 *	0.019	0.958 (0.923–0.994)
SIOPS										0.050	0.028	1.052 (0.995–1.111)
Intercept	0.472 ***	0.075	1.603 (0.844–1.246)	0.025	0.099	1.026 (0.976–1.460)	0.177	0.103	1.194 (0.844–1.246)	0.437	0.294	1.549 (0.870–2.756)
*n*	29177			28611			28611			7400		
Log Likelihood	−15988.8			−15606.8			−15519.1			−4334.4		
BIC	32111.2			31388.1			31243.5			8900.4		

Note: * *p* < 0.05, ** *p* < 0.01, *** *p* < 0.001. NCD: non-communicable disease; SRH: self-rated health; UEBMI: Urban Employee Basic Medical Insurance; URBMI: Urban Resident Basic Medical Insurance; NCMS: New Cooperative Medical Scheme; GIS: Government Employee Insurance Scheme; SMI: Supplementary Medical Insurance; SIOPS: Standard International Occupational Prestige Scale; BIC: Bayesian Information Criterion; SE: standard error; OR: odds ratio; CI: confidence interval.

**Table 3 ijerph-17-04131-t003:** Logistic regression of seeking medical treatment if sick within 2 weeks.

Variable	Model-5 (2010)			Model-6 (2012)			Model-7 (2014)			Model-8 (2016)		
	B	SE	OR (95%CI)	B	SE	OR (95%CI)	B	SE	OR (95%CI)	B	SE	OR (95%CI)
Gender (Ref. = Female)	−0.016	0.100	0.984 (0.808–1.198)	0.036	0.115	1.036 (0.827–1.299)	−0.116	0.096	0.891 (0.738–1.074)	−0.118	0.167	0.888 (0.640–1.233)
Age	0.003	0.005	1.003 (0.993–1.013)	−0.008	0.006	0.992 (0.980–1.003)	0.016 **	0.005	1.016 (1.006–1.026)	0.033 ***	0.009	1.034 (1.015–1.052)
Household Registration (Ref. = Agricultural)												
Non-Agricultural	0.411	0.233	1.508 (0.954–2.383)	−0.114	0.208	0.892 (0.593–1.342)	−0.040	0.168	0.961 (0.691–1.334)	0.095	0.239	1.099 (0.687–1.757)
Others	0.000	(empty)	(empty)				−0.774	1.124	0.461 (0.050–4.175)	0.000	(empty)	(empty)
Marital Status (Ref. = Never Married)												
Married	0.546 *	0.243	1.727 (1.073–2.779)	0.736 **	0.249	2.088 (1.281–3.400)	0.057	0.207	1.059 (0.706–1.589)	0.027	0.351	1.027 (0.516–2.042)
Others	0.815 *	0.352	2.259 (1.134–4.500)	0.462	0.355	1.587 (0.791–3.181)	0.250	0.273	1.284 (0.753–2.191)	0.437	0.540	1.548 (0.537–4.463)
Retired (Ref. = no)	−0.473	0.368	0.623 (0.303–1.282)	0.721 *	0.300	2.057 (1.143–3.702)	−0.108	0.147	0.898 (0.674–1.197)	0.261	0.466	1.299 (0.521–3.237)
NCD (Ref. = no)	0.783 ***	0.118	2.188 (1.736–2.757)	0.681 ***	0.147	1.976 (1.481–2.637)	1.051 ***	0.117	2.860 (2.272–3.599)	1.192 ***	0.221	3.293 (2.135–5.178)
SRH (Ref.= Very Unhealthy)												
Unhealthy	−0.956 *	0.453	0.384 (0.158–0.934)	−0.757 ***	0.152	0.469 (0.348–0.632)	−0.588 ***	0.142	0.556 (0.426–0.733)	−0.440	0.265	0.644 (0.383–1.082)
Relatively Unhealthy	−1.020 *	0.459	0.361 (0.167–0.886)	−0.959	0.146	0.383 (0.288–0.511)	−0.619 ***	0.130	0.539 (0.418–0.695)	−0.590 *	0.245	0.555 (0.343–0.896)
Fair	−1.187 **	0.442	0.305 (0.128–0.725)	−1.025 ***	0.223	0.359 (0.232–0.555)	−0.620 ***	0.187	0.538 (0.373–0.776)	−1.175 ***	0.332	0.309 (0.161–0.592)
Healthy	−1.363 **	0.448	0.256 (0.106–0.615)	−0.961 **	0.337	0.382 (0.198–0.739)	−0.492 *	0.229	0.611 (0.389–0.958)	−0.592	0.395	0.553 (0.255–1.201)
Social Medical Insurance (Ref.= UEBMI)												
URBMI	−0.060	0.245	0.942 (0.583–1.521)	−0.492 *	0.231	0.611 (0.389–0.961)	0.188	0.196	1.207 (0.821–1.773)	0.155	0.290	1.167 (0.661–2.059)
NCMS	0.459	0.256	1.583 (0.958–2.615)	0.181	0.232	1.198 (0.761–1.887)	0.474 *	0.186	1.606 (1.116–2.313)	0.576 *	0.260	1.779 (1.068–2.961)
PMI	0.318	0.235	1.374 (0.886–2.179)	−0.119	0.257	0.888 (0.537–1.467)	0.045	0.251	1.046 (0.639–1.711)	0.309	0.396	1.362 (0.626–2.962)
SMI	0.000	0.327	1.000 (0.527–1.897)	−1.512 *	0.604	0.221 (0.067–0.721)	−0.025	0.328	0.975 (0.512–1.856)	−0.745	0.476	0.475 (0.187–1.207)
Area (Ref. = West)												
Center	−0.256 *	0.130	0.774 (0.599–0.998)	−0.128	0.149	0.88 (0.657–1.178)	−0.019	0.133	0.981 (0.756–1.272)	−0.076	0.236	0.927 (0.583–1.473)
East	−0.047	0.125	0.954 (0.747–1.218)	0.043	0.141	1.044 (0.792–1.375)	−0.143	0.120	0.867 (0.686–1.096)	−0.134	0.212	0.875 (0.577–1.326)
North-East	−0.547 ***	0.157	0.579 (0.425–1.787)	−0.633 ***	0.176	0.531 (0.376–0.750)	−0.694 ***	0.147	0.500 (0.375–0.666)	−0.589 *	0.273	0.555 (0.325–0.948)
Education (Ref. = Primary School or Below)												
Middle School	0.016	0.122	1.016 (0.799–1.291)	0.025	0.136	1.025 (0.786–1.337)	0.065	0.126	1.067 (0.834–1.365)	−0.087	0.224	0.916 (0.591–1.420)
High School	−0.302	0.166	0.74 (0.534–1.025)	−0.192	0.186	0.826 (0.574–1.188)	−0.146	0.161	0.864 (0.630–1.185)	−0.397	0.273	0.672 (0.394–1.148)
Bachelor’s Degree	−0.989 ***	0.246	0.372 (0.229–0.603)	0.042	0.252	1.043 (0.636–1.709)	−0.220	0.210	0.803 (0.532–1.211)	−0.161	0.313	0.851 (0.461–1.573)
Master’s Degree	−0.122	0.791	0.885 (0.188–4.169)	0.000	(empty)	(empty)	0.147	0.751	1.159 (0.265–5.253)	−0.190	0.762	0.827 (0.186–3.684)
Personal Annual Income	−0.010	0.040	0.990 (0.916–1.069)	−0.046	0.050	0.955 (0.865–1.653)	−0.045	0.029	0.956 (0.902–1.012)	0.006	0.104	1.006 (0.820–1.233)
SIOPS	0.067	0.006	1.007 (0.958–1.195)	−0.030	0.006	0.997 (0.861–1.093)	0.084	0.005	1.008 (0.427–3.143)	0.038	0.006	1.004 (0.920–1.173)
Intercept	0.677	0.706	1.968 (0.493–7.855)	1.292	0.675	3.64 (0.968–13.677)	0.142	0.512	1.153 (0.422–3.143)	−0.782	1.223	0.457 (0.415–5.034)
*n*	2244			1684			2612			855		
Log Likelihood	−1328.8			−1040.2			−1413			−491.8		
BIC	2850.5			2258.7			3030.6			1152.5		

Note: * *p* < 0.05, ** *p* < 0.01, *** *p* < 0.001. NCD: non-communicable disease; SRH: self-rated health; UEBMI: Urban Employee Basic Medical Insurance; URBMI: Urban Resident Basic Medical Insurance; NCMS: New Cooperative Medical Scheme; GIS: Government Employee Insurance Scheme; SMI: Supplementary Medical Insurance; SIOPS: Standard International Occupational Prestige Scale; BIC: Bayesian Information Criterion; SE: standard error; OR: odds ratio; CI: confidence interval.

**Table 4 ijerph-17-04131-t004:** Longitudinal analysis (2010−2016) of a specialist visiting.

Variable	Model-9			Model-10			Model-11			Model-12		
	B	SE	OR (95%CI)	B	SE	OR (95%CI)	B	SE	OR (95%CI)	B	SE	OR (95%CI)
Gender (Ref. = Female)	0.018	0.020	1.018 (0.978–1.059)	−0.023	0.020	0.977 (0.538–1.017)	−0.016	0.020	0.984 (0.945–1.024)	−0.191 ***	0.039	0.826 (0.765–0.892)
Age	−0.011 ***	0.001	0.989 (0.987–0.990)	−0.011 ***	0.001	0.990 (0.988–0.991)	−0.011 ***	0.001	0.989 (0.987–0.991)	−0.005 *	0.002	0.995 (0.991–0.999)
Household registration (Ref. = Agricultural)												
Non-Agricultural	1.627 ***	0.022	5.090 (4.870–5.318)	0.749 ***	0.033	2.116 (1.985–2.255)	0.730 ***	0.033	2.074 (0.994–2.213)	0.679 ***	0.057	1.971 (1.763–2.203)
Others	0.903 ***	0.271	2.467 (1.450–4.195)	0.479 *	0.233	1.614 (1.021–2.549)	0.402	0.238	1.494 (0.937–2.381)	0.596	0.405	1.815 (0.820–4.014)
Marital Status (Ref. = Never married)												
Married	−0.061	0.038	0.941 (0.873–1.013)	−0.071	0.038	0.932 (0.865–1.004)	−0.078 *	0.038	0.925 (0.858–0.997)	−0.196 **	0.070	0.822 (0.716–0.942)
Others	−0.027	0.055	0.973 (0.874–1.084)	−0.034	0.055	0.967 (0.867–1.077)	−0.063	0.056	0.939 (0.842–1.046)	−0.100	0.113	0.904 (0.725–1.127)
Retired (Ref. = no)	0.227 ***	0.028	1.255 (1.187–1.327)	0.152 ***	0.029	1.165 (1.099–1.233)	0.155 ***	0.030	1.168 (1.101–1.238)	0.302 ***	0.074	1.352 (1.169–1.562)
NCD (Ref. = no)	0.474***	0.023	1.606 (1.534–1.681)	0.468 ***	0.024	1.596 (1.522–1.672)	0.491 ***	0.024	1.635 (1.559–1.714)	0.468 ***	0.051	1.597 (1.444–1.765)
SRH (Ref.= Very Unhealthy)												
Unhealthy	−0.422 ***	0.029	0.656 (0.619–0.695)	−0.466 ***	0.030	0.628 (0.592–0.666)	−0.436 ***	0.030	0.647 (0.609–0.686)	−0.619 ***	0.071	0.539 (0.469–0.618)
Relatively Unhealthy	−0.387 ***	0.027	0.679 (0.644–0.716)	−0.466 ***	0.028	0.627 (0.594–0.662)	−0.454 ***	0.028	0.635 (0.601–0.671)	−0.665 ***	0.065	0.514 (0.452–0.584)
Fair	−0.739 ***	0.031	0.478 (0.449–0.508)	−0.781 ***	0.032	0.458 (0.431–0.487)	−0.766 ***	0.032	0.465 (0.436–0.495)	−0.979 ***	0.070	0.376 (0.327–0.431)
Healthy	−0.720 ***	0.035	0.487 (0.455–0.521)	−0.734 ***	0.035	0.480 (0.447–0.514)	−0.726 ***	0.036	0.484 (0.451–0.519)	−0.929 ***	0.073	0.395 (0.342–0.456)
Social Medical Insurance (Ref.= UEBMI)												
URBMI				−0.499 ***	0.038	0.607 (0.563–0.654)	−0.482 ***	0.038	0.618 (0.573–0.665)	−0.287 ***	0.065	0.751 (0.660–0.853)
NCMS				−1.341 ***	0.037	0.262 (0.243–0.281)	−1.299 ***	0.037	0.273 (0.253–0.294)	−0.870 ***	0.062	0.419 (0.371–0.473)
GIS				0.045	0.052	1.046 (0.943–1.159)	0.050	0.053	1.051 (0.947–1.165)	−0.104	0.083	0.901 (0.766–1.059)
SMI				−0.329 ***	0.073	0.719 (0.623–0.830)	−0.310 ***	0.074	0.734 (0.634–0.848)	0.050	0.114	1.051 (0.840–1.314)
Area (Ref. = West)												
Center							−0.117 ***	0.028	0.890 (0.841–0.940)	−0.248 ***	0.054	0.780 (0.701–0.868)
East							0.020	0.026	1.021 (0.969–1.074)	−0.010	0.048	0.99 (0.900–1.086)
North-East							0.679 ***	0.033	1.972 (1.850–2.101)	0.717 ***	0.062	2.049 (1.815–2.313)
Education (Ref. = Primary School or Below)												
Middle School										0.204 ***	0.048	1.226 (1.116–1.347)
High School										0.481 ***	0.059	1.618 (1.439–1.818)
Bachelor’s Degree										0.787 ***	0.076	2.196 (1.894–2.547)
Master’s Degree										1.263 ***	0.342	3.535 (1.809–6.906)
Personal Annual Income										0.157 ***	0.015	1.171 (1.137–1.205)
SIOPS										0.02	0.017	1.002 (0.990–1.059)
Intercept	=0.346 ***	0.048	0.708 (0.644–0.778)	0.974 ***	0.059	2.649 (2.355–2.979)	0.895 ***	0.062	2.447 (2.167–2.763)	−1.114 ***	0.207	0.328 (0.218–0.493)
*n*	80495			79589			79583			21722		
Log Likelihood	−44821.8			−43401.4			−42957.9			−11352		
BIC	89790.5			86994.7			86141.5			22963.6		

Note: * *p* < 0.05, ** *p* < 0.01, *** *p* < 0.001. Note: NCD: non-communicable disease; SRH: self-rated health; UEBMI: Urban Employee Basic Medical Insurance; URBMI: Urban Resident Basic Medical Insurance; NCMS: New Cooperative Medical Scheme; GIS: Government Employee Insurance Scheme; SMI: Supplementary Medical Insurance; SIOPS: Standard International Occupational Prestige Scale; BIC: Bayesian Information Criterion; SE: standard error; OR: odds ratio; CI: confidence interval.

**Table 5 ijerph-17-04131-t005:** Logistic regression of a specialist visiting for the insured.

Variable	Model-13 (2010)			Model-14 (2012)			Model-15 (2014)			Model-16 (2016)		
	B	SE	OR (95%CI)	B	SE	OR (95%CI)	B	SE	OR (95%CI)	B	SE	OR (95%CI)
Gender (Ref. = Female)	−0.351 **	0.135	0.704 (0.540–0.918)	−0.291 ***	0.070	0.748 (0.652–0.857)	−0.150 **	0.053	0.860 (0.775–0.955)	−0.219 **	0.078	0.803 (0.689–0.935)
Age	−0.037 ***	0.007	0.964 (0.951–0.977)	0.003	0.004	1.003 (0.995–1.009)	−0.006 *	0.003	0.994 (0.988–0.999)	−0.004	0.004	0.996 (0.987–1.003)
Household Registration (Ref. = Agricultural)												
Non-Agricultural	1.176 ***	0.252	3.240 (1.976–5.310)	0.710 ***	0.105	2.034 (1.655–2.498)	0.632 ***	0.080	1.881 (1.606–2.201)	0.665 ***	0.099	1.945 (1.603–2.359)
Others	0.000	(empty)	(empty)	0.000	(empty)	(empty)	0.430	0.786	1.537 (0.329–7.154)	0.505	0.890	1.657 (0.289–2.478)
Marital Status (Ref. = Never Married)												
Married	−0.276	0.245	0.758 (0.469–1.225)	−0.251	0.129	0.778 (0.604–1.001)	0.029	0.097	1.029 (0.851–1.243)	−0.191	0.145	0.826 (0.621–1.098)
Others	0.290	0.465	1.337 (0.537–3.328)	−0.012	0.215	0.988 (0.648–1.505)	−0.011	0.148	0.989 (0.740–1.321)	−0.241	0.230	0.786 (0.500–1.232)
Retired (Ref. = no)	1.191 **	0.449	3.291 (1.365–7.930)	−0.258	0.185	0.772 (0.537–1.110)	−0.020	0.087	0.98 (0.826–1.162)	0.218	0.192	1.244 (0.853–1.813)
NCD (Ref. = no)	0.289	0.187	1.335 (0.925–1.924)	0.575 ***	0.099	1.777 (1.463–2.156)	0.590 ***	0.073	1.804 (1.546–2.079)	0.536 ***	0.120	1.709 (1.349–2.164)
SRH (Ref.= Very Unhealthy)												
Unhealthy	0.013	0.291	1.013 (0.572–1.790)	−0.532 ***	0.118	0.587 (0.466–0.793)	−0.504 ***	0.098	0.604 (0.498–0.732)	−0.504 **	0.165	0.604 (0.436–0.835)
Relatively Unhealthy	−0.319	0.325	0.727 (0.384–1.374)	−0.595 ***	0.109	0.551 (0.445–0.682)	−0.665 ***	0.088	0.514 (0.433–0.610)	−0.454 **	0.155	0.635 (0.469–0.859)
Fair	−0.086	0.138	0.917 (0.699–1.203)	−0.877 ***	0.125	0.416 (0.325–0.531)	−0.641 ***	0.099	0.527 (0.433–0.639)	−0.707 ***	0.168	0.493 (0.354–0.685)
Healthy	0.000	(omitted)	(omitted)	−0.744 ***	0.146	0.475 (0.356–0.633)	−0.608 ***	0.107	0.545 (0.442–0.671)	−0.403 *	0.173	0.668 (0.475–0.937)
Social Medical Insurance (Ref.= UEBMI)												
URBMI	0.009	0.261	1.009 (0.605–1.682)	−0.159	0.124	0.853 (0.669–0.087)	−0.376 ***	0.096	0.687 (0.568–0.828)	−0.300 *	0.125	0.741 (0.580–0.946)
NCMS	−0.991 ***	0.286	0.371 (0.211–0.649)	−0.746 ***	0.117	0.474 (0.376–0.596)	−0.838 ***	0.090	0.432 (0.362–0.516)	−0.792 ***	0.106	0.453 (0.368–0.557)
GIS	0.014	0.270	1.014 (0.597–1.722)	0.153	0.147	1.165 (0.873–1.555)	−0.084	0.133	0.919 (0.708–1.192)	0.012	0.177	1.012 (0.715–1.431)
SMI	0.051	0.444	1.052 (0.440–2.514)	0.020	0.285	1.021 (0.583–1.785)	−0.130	0.179	0.878 (0.618–1.246)	0.454	0.235	1.575 (0.993–2.496)
Area (Ref. = West)												
Center	−1.218 ***	0.195	0.296 (0.201–0.433)	−0.318 **	0.097	0.728 (0.601–0.879)	−0.133	0.073	0.875 (0.758–1.010)	−0.169	0.107	0.844 (0.684–1.042)
East	−0.758 ***	0.160	0.469 (0.342–0.641)	0.004	0.085	1.004 (0.849–1.186)	0.068	0.067	1.07 (0.939–1.219)	−0.007	0.096	0.993 (0.822–1.197)
North-East	−0.429	0.314	0.651 (0.352–1.203)	0.660 ***	0.107	1.935 (1.569–2.386)	0.711 ***	0.084	2.037 (1.727–2.401)	0.773 ***	0.127	2.167 (1.688–2.781)
Education (Ref. = Primary School or Below)												
Middle School	−0.442 **	0.167	0.643 (0.463–0.892)	0.169 *	0.084	1.185 (1.004–1.397)	0.268 ***	0.068	1.307 (1.144–1.492)	0.242 *	0.098	1.274 (1.052–1.542)
High School	−0.268	0.231	0.765 (0.486–1.203)	0.504 ***	0.104	1.656 (1.349–2.031)	0.552 ***	0.084	1.737 (1.473–2.048)	0.337 **	0.116	1.400 (1.114–1.758)
Bachelor’s Degree	−0.259	0.287	0.772 (0.439–1.354)	0.750 ***	0.137	2.117 (1.618–2.769)	0.835 ***	0.108	2.304 (1.863–2.848)	0.646 ***	0.138	1.909 (1.455–2.502)
Master’s Degree	0.175	1.090	1.192 (0.140–10.093)	2.994 **	1.051	19.964 (2.543–156.667)	1.064 *	0.485	2.899 (1.120–7.495)	0.599	0.452	1.821 (0.750–4.419)
Personal Annual Income	0.313 ***	0.071	1.368 (1.191–1.571)	0.181 ***	0.033	1.198 (1.122–1.279)	0.081 ***	0.017	1.085 (1.048–1.122)	0.274 ***	0.047	1.315 (1.199–1.441)
SIOPS	0.058	0.064	1.060 (0.934–1.201)	−0.015	0.035	0.985 (0.920–1.054)	0.036	0.025	1.037 (0.987–1.089)	−0.219 **	0.078	1.087 (1.028–1.148)
Intercept	−2.939 **	0.777	0.131 (0.028–0.602)	−1.724 ***	0.412	0.178 (0.079–0.399)	−0.626 *	0.276	0.535 (0.311–0.918)	−2.527 ***	0.547	0.080 (0.027–0.233)
*n*	3137			5801			8622			4128		
Log Likelihood	−878.4			−2992.6			−4777.5			−2382.3		
BIC	1950			6201.9			9790.6			4981		

Note: * *p* < 0.05, ** *p* < 0.01, *** *p* < 0.001. Note: NCD: non-communicable disease; SRH: self-rated health; UEBMI: Urban Employee Basic Medical Insurance; URBMI: Urban Resident Basic Medical Insurance; NCMS: New Cooperative Medical Scheme; GIS: Government Employee Insurance Scheme; SMI: Supplementary Medical Insurance; SIOPS: Standard International Occupational Prestige Scale; BIC: Bayesian Information Criterion; SE: standard error; OR: odds ratio; CI: confidence interval.

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
