# Peer review of "Which Matters for Medical Utilization Equity under Universal Coverage: Insurance System, Region or SES"

_ijerph, 2020, doi:10.3390/ijerph17114131_

Round 1

Reviewer 1 Report

  • There is a different number format on lines 99 and later on lines 101 and 102. It refers to the punctuation of thousands. Later it happens the same in the tables. I don't know what the recommended format is, but it's easier to read the numbers when the thousands are marked. If the recommendation is different, do not consider this recommendation.
  • NCD first appears on line 78. Later it appears on line 129. It is not understood what it refers to.
  • SRH must be related to health status, but is not defined. Appears on line 129.
  • On line 156 we can read OR, also in the Abstract. I can’t find it’s meaning previously. If I am wrong, please disregard this comment.
  • In line 233 it is not understood what OPP refers to.
  • It can be read in lines 234 and 235: “… ability to access to specialist visiting especially for inpatient utilization especially for the poor rural residents.” It appears twice especially.
  • On page 242 a dot must follow results.
  • You should review the format of the References. Pages are missing in some of them, the years of the publications are not always in bold. In reference 62 the title of the magazine is not complete in italics. Reference 44 has two points on the pages, etc.

Reviewer 2 Report

The study investigated the medical insurance utilization disparities over different insurance schemes, regions and SES, by use of a longitudinal survey of Chinese communities, families, and individuals. Below are a few comments that may help improve the manuscript.

  1. This manuscript would be read over and edited extensively by a native English speaker to bring it to the level where it can be considered for publication.
  2. UEBMI, URBMI, NCMS, and SIOPS should be spelled out in the abstract.
  3. The authors mentioned that universal coverage of medical insurance has been launched in China. However, according to their descriptions, people live in China are included in different medical insurance schemes. The authors might need to be careful about the term, “universal coverage,” which they used in the manuscript.
  4. What is “SMI”, “NCD”, “SRH”?
  5. I would recommend a careful proofread of the draft to improve consistency. For example, the authors used “NCMS” and “UEMI” in the whole manuscript. However, they mentioned “NRCMS” and UEMBI” in the introduction.
  6. The authors described “household registration (1=agricultural; 2=non-agricultural; 3=others), …” [page 3, line 132]. However, in table 1, household registration includes urban, rural, and others. 
  7. In 2012 and 2016, Why are the number of people with NRCMS higher than the number of people who register in a rural area?
  8. The authors might briefly introduce the index of SIOPS such as how to calculate and interpret the index. 
  9. What assumptions did the authors apply for the time effects when conducting longitudinal analysis?
  10. Did the authors test collinearity in models especially for education, annual income, and SIOPS?
  11. The authors may reveal information on missing values. The sample size decreased dramatically in models 4 [28611 to 12332] and 12 [79583 to 21772], which needs to be discussed in the manuscript.
  12. What did the unit be analyzed in models 5-8 and 13-16? The sample sizes are much smaller than each wave the authors mentioned, which needs to clarify.
  13. What is the reference group in gender in tables 3, 4, and 5?
  14. The abbreviations showed in tables should be listed in the footnote.
  15. Limitations of this study should be mentioned.
  16. Do different insurance schemes execute any regulations to limit people’s seeking behavior?
  17. The sentence is not clear “However, separate models in this study suggested that the disparities between NCMS and UEBMI had decreased over years.” [page 8, line 231]. Did the authors conduct a trend test in the study?
  18. Does “OOP” mean “out of pocket”? The full name should be spelled out.

Reviewer 3 Report

Please check the attached file with memo.

At first, You have to explain more detail regarding many models. I couldn't get any information regarding the difference among models clearly.

Author Response

plsease see the attachment

Reviewer 4 Report

This study utilized data from the China Family Panel Studies to assess whether medical insurance utilization after universal coverage is equitable across insurance systems, region and socioeconomic status. The authors reported inequalities by the three factors under study indicating that members of the new rural cooperative medical system insurance and those residing in the Western region were more likely to seek medical treatment if ill, though not likely to visit specialists. The authors also reported a positive correlation between higher socioeconomic status and increased medical utilization.  

  1. Statistical methods are unclear and thus difficult to review. It seems the authors used Hierarchical models? If so, please describe in detail, the statistical analysis. What do the numbers in the tables refer to? I assume these are the parameter estimates from the models? Odds ratios? Please clearly label the tables and add standard errors [g. estimated coefficient (SE)]. Were fixed effects or random effects used?
  2. Authors state that survey samples are representative of the population, did authors check if insured population within the surveys is also representative of insured target population?
  3. Throughout the paper, the authors refer to odds ratios as percentages, which is incorrect. Odds may be higher or lower when comparing groups, not odds ratios. Recommend reporting the odds ratio and 95% confidence interval in the text.
  4. Recommend commenting on any design changes or data collection changes over the years of the surveys, whether or not there were any. This could be added to the discussion section; recommend authors address all known limitations of the study in the discussion.
  5. Please define acronyms in the abstract at first appearance. Also, the acronym NCMS was never defined? I assumed this was the same as NRCMS? If so, please be consistent.
  6. For SIOP, consider reporting change per 5 or 10 unit increase to be able to report a > 0% difference.

Round 2

Reviewer 2 Report

Generally, I am satisfied with the revision that has been improved significantly. I do not have more comments. 

Author Response

Dear Reviewer,

We are so glad to hear from you  soon! We really appreciate all your suggestions which have helped us to revise our paper substantially. And to myself, I have learned from your suggestions, and which have inspired me to do some new research!  Sincerely, thanks for all your advice!

Best,

Joy Huang

jiaoling_huang@sina.com

Reviewer 3 Report

I recommend to publish the manuscript

Author Response

Dear reviewers,

We are so excited to hear from you so soon! I learned a lot from your suggestions and which have inspired me to explore some new area! We appreciate all your advice to our paper, and sincerely, thanks a lot to help me keep moving in doing research! 

Best,

Joy Huang

jiaoling_huang@sina.com

Reviewer 4 Report

  1. Recommend adding odds ratio in the parenthesis with the 95% confidence interval to avoid confusion. Reporting the odds, then a 95% confidence interval of the odds ratio without reporting the actual estimate for which the confidence interval is referring to may be confusing. 
  2. Make sure your tables can stand alone and that you indicate what you are reporting. (e.g., Table 1, indicate that this is N (%). 
